# Molecular mechanism of bacteriophage contraction structure of an S-layer–penetrating bacteriophage

Jason S Wilson[1] , Louis-Charles Fortier[2] , Robert P Fagan[1,3] , Per A Bullough[1,3]

The molecular details of phage tail contraction and bacterial cell envelope penetration remain poorly understood and are completely unknown for phages infecting bacteria enveloped by proteinaceous S-layers. Here, we reveal the extended and contracted atomic structures of an intact contractile-tailed phage (φCD508) that binds to and penetrates the protective S-layer of the Gram-positive human pathogen *Clostridioides difficile*. The tail is unusually long (225 nm), and it is also notable that the tail contracts less than those studied in related contractile injection systems such as the model phage T4 (~20% compared with ~50%). Surprisingly, we find no evidence of auxiliary enzymatic domains that other phages exploit in cell wall penetration, suggesting that sufficient energy is released upon tail contraction to penetrate the S-layer and the thick cell wall without enzymatic activity. Instead, the unusually long tail length, which becomes more flexible upon contraction, likely contributes toward the required free energy release for envelope penetration.

## Introduction

Phages are the most abundant biological entities on earth, yet only a tiny fraction has been characterized in any detail (Hatfull, 2015). The most prevalent phage morphology described is that of an icosahedral capsid, containing the genome, attached to a tail (Ackermann, 2007). The tail forms a conduit through which the genome is delivered into the host cell cytoplasm and is also associated with receptor binding functions (Hardy et al, 2022). Among tailed phages, myoviruses have contractile tails of widely varying lengths, podoviruses have short noncontractile tails, whereas siphoviruses have long, flexible, noncontractile tails (Fokine & Rossmann, 2014).

Myovirus phage tails and bacterial tailocins both belong to the group of so-called contractile injection systems (CISs) (Taylor et al, 2018). The most extensively studied myovirus structure is that of the Gram-negative *Escherichia coli* phage, T4 (Fokine et al, 2004, 2013;

Leiman et al, 2010; Sun et al, 2015; Taylor et al, 2016; Chen et al, 2017). The T4 contractile tail shares many features in common with tailocins, although tailocins lack a capsid and do not inject DNA into their target and instead inject effector proteins (Yang et al, 2006; Quentin et al, 2018; Desfosses et al, 2019; Jiang et al, 2019). For CISs, most detailed structural information only covers the initially extended tail conformation. However, contracted state structures have been reported at high resolution for *Pseudomonas* phage E217, and a tailocin, R2 pyocin, the type VI secretion system (T6SS), a *Clostridioides difficile* bacteriocin, and the *Staphylococcus aureus* phage phi812 (Kudryashev et al, 2015; Wang et al, 2017; Ge et al, 2020; Li et al, 2023; Biňovský et al, 2024 *Preprint*; Cai et al, 2024).

Although other known myovirus structures share common tail features with T4, and E217 and phi812, the molecular interactions stabilizing their contracted tail state are unknown (Novacek et al, 2016; Guerrero-Ferreira et al, 2019; Wang et al, 2023; Yang et al, 2023). Moreover, it is not known to what extent the subunit movements within the tail during contraction are the same across myoviruses or CISs in general. Although phages target bacteria with a wide variety of cell envelope types, little is known about the penetration of the thick Gram-positive peptidoglycan cell wall, although endopeptidase and hydrolase domains within the baseplate and needle proteins are common adaptations (Latka et al, 2017; Dunne et al, 2018). These small lytic domains contain conserved folds that make them readily identifiable from sequence and tertiary structure. It is notable, however, that T6SS assemblies appear to have no such lytic domains and to rely entirely on mechanical force to penetrate target cell envelopes (Brackmann et al, 2017; Wang et al, 2017). Even less is known about infection through the additional protective proteinaceous S-layer found in most of the eubacteria (Fagan & Fairweather, 2014). An important unanswered question is whether S-layer–penetrating contractile phages have structural features that distinguish them from other phages.

*Clostridioides difficile* infection is the most common cause of antibiotic-associated diarrhea, resulting in significant morbidity and mortality in susceptible populations (Guery et al, 2019). In common with many bacterial species, *C. difficile* produces a proteinaceous S-layer that forms the outermost surface of the cell envelope and acts as the receptor for most of the phages that infect

[1]Molecular Microbiology, School of Biosciences, University of Sheffield, Sheffield, UK  [2]Department of Microbiology and Infectious Diseases, Faculty of Medicine and Health Sciences, Université de Sherbrooke, Sherbrooke, Canada  [3]The Florey Institute, University of Sheffield, Sheffield, UK

Correspondence: j.s.wilson@sheffield.ac.uk; r.fagan@sheffield.ac.uk; p.bullough@sheffield.ac.uk

the species (Kirk et al, 2017; Lanzoni-Mangutchi et al, 2022; Royer et al, 2023). Phages have been proposed as alternative antimicrobials to treat *C. difficile* infection (Kortright et al, 2019); naturally occurring phages can be used as a template for the design of such precision antimicrobials (Gebhart et al, 2015; Kirk et al, 2017). To advance designs, we need to describe the essential structural components and the mechanics involved in binding to the host cell and penetration of the host cell envelope. *φ*CD508 is a lysogenic contractile-tailed phage in the genus *Colneyvirus* that infects strains expressing the S-layer cassette type (SLCT)-10 (Sekulovic et al, 2014; Royer et al, 2023). To date, all *C. difficile*–infecting phages that have been identified and characterized are lysogenic, likely an adaptation to the life cycle of a sporulating anaerobic host (Hargreaves & Clokie, 2014). Identification or engineering of a strictly lytic phage would be preferable for future therapeutic applications (Selle et al, 2020), but lysogenic phages have shown some success in hamsters and in a human gut microbiome model (Nale et al, 2016, 2018).

Here, we describe the high-resolution structure of the fully intact *φ*CD508 virion in both its extended and contracted form. We observe that the extent of contraction is significantly less than that reported for other phage structures. This has important implications for the mechanism of S-layer–penetrating phages in general. We also highlight key differences in previously characterized myoviruses: the baseplate of the tail is less complex; the needle spike is much more compact than that of other CISs and remarkably appears to lack enzymatic domains.

## Results

### Overall structure of *φ*CD508

*φ*CD508 is a myovirus (Sekulovic et al, 2014), with an overall tail length of 225 nm. The 49,272-bp *φ*CD508 genome encodes 74 predicted gene products, 26 of which comprise the structural cassette (Supplemental Data 1). Tandem mass spectrometry of purified phage virions detected 22 of these 26 proteins, missing were two small proteins of unknown function (gp52 and gp58), a putative XkdN-like tail assembly chaperone gp57, and a putative membrane protein of unknown function gp60. Among the 22 proteins detected were 17 putative structural proteins, the capsid scaffold gp46, capsid protease gp47, putative chaperones gp69 and gp70, and a small protein of unknown function gp54.

Structures of purified *φ*CD508 virions in both extended and contracted conformations were determined by single-particle EM of frozen-hydrated samples (Fig 1A and C); the resolution ranged from 2.6 to 4.0 Å for the extended phage (Fig S1A), and 2.9 to 4.2 Å for the contracted form (Table S1, Fig S1B). In the extended state, 14 of the 17 predicted structural proteins identified by mass spectrometry (Supplemental Data 1) were revealed and modeled at or close to their full chain length. Missing from the final structure are the tail fiber (gp67) and predicted receptor binding protein (gp68); only a short section of the tape measure protein (gp59) was modeled. Each protein chain was built ab initio where the resolution was sufficient, or was first modeled by RoseTTAFold (Baek et al, 2021) or AlphaFold

(Jumper et al, 2021) and then fitted into the maps of both the extended and the contracted phage (Fig 1B). The virion can be divided into five distinct regions: the head, neck, tail, baseplate, and needle (Fig 1A).

### Head

The 650-Å-diameter head (Fig 2A) is formed of a near-complete T = 7 icosahedral capsid (Caspar & Klug, 1962) and houses the genomic DNA. It is made up of 415 and 420 copies of two capsomer proteins, the major capsid protein gp49 and the capsid decoration protein gp48, respectively (Fig S2A–F). Within the capsid of the extended phage, internal density shows layers of packaged DNA (Fig 2A). The contracted virion capsid reconstruction contains an identical arrangement of the major capsid protein and capsid decoration protein but lacks internal density indicating that the DNA has been ejected from the head. At the base of the capsid sits a unique opening, centered on what would be a fivefold symmetry axis in a complete icosahedron; it is made up of 12 copies of portal protein gp45 (Figs 2B and S2G and H), forming an opening through which DNA is ejected from the capsid. Thus, there is a 5–12-fold symmetry mismatch between the capsid and the portal (Fig S2J and K; Supplemental Data 2).

### Neck

The portal in turn connects to the neck (Figs 2C and S2I), which is made up of three protein complexes with a VIRFAM type I neck arrangement (Lopes et al, 2014), comprising one dodecameric ring (gp50) and two hexameric rings (gp51 and gp53) (Fig 2C). The symmetry is therefore reduced from 12-fold down to 6-fold, traveling down the neck. The neck links the capsid to the tail (Fig 2C). The first complex in the neck is the head-to-tail adaptor gp50, which assembles into a 12-fold symmetric ring and acts to resolve a symmetry mismatch between the portal and the tail (Figs 2C and S3A–C, E, and F). The neck valve protein gp51 (Fig S3D) forms a hexameric complex and represents the narrowest constriction in the neck lumen, ~23 Å in diameter at its minimum (Fig S3E). gp51 has structural homology to the "stopper" protein of SPP1, which is proposed to prevent DNA leakage from the capsid (Table S2) (Lhuillier et al, 2009) (RMSD ~8 Å for 53 out of 80 Cα pairs). Other similar structures identified by Foldseek (van Kempen et al, 2024) include the Pam3 connector protein (Yang et al, 2023), E217 collar protein gp28 (Li et al, 2023), XM1 head completion protein (Wang et al, 2023), stopper protein of gene transfer agent (Bardy et al, 2020), and neck 2 protein gp15 of Milano (Sonani et al, 2023), where RMSD values range from ~4 to ~6 Å for ~95 Cα pairs. In our structure, strong density in the lumen of the channel weakens around the neck valve (Fig S2I), suggesting the neck valve may also constitute the point at which DNA is prevented from exiting through the tail of the *φ*CD508 before contraction. Below the constriction, a weaker density is present, which likely constitutes the tape measure protein gp59, although the enforced sixfold symmetry in the reconstruction prevents identification or building of a model into this region (Fig S2I).

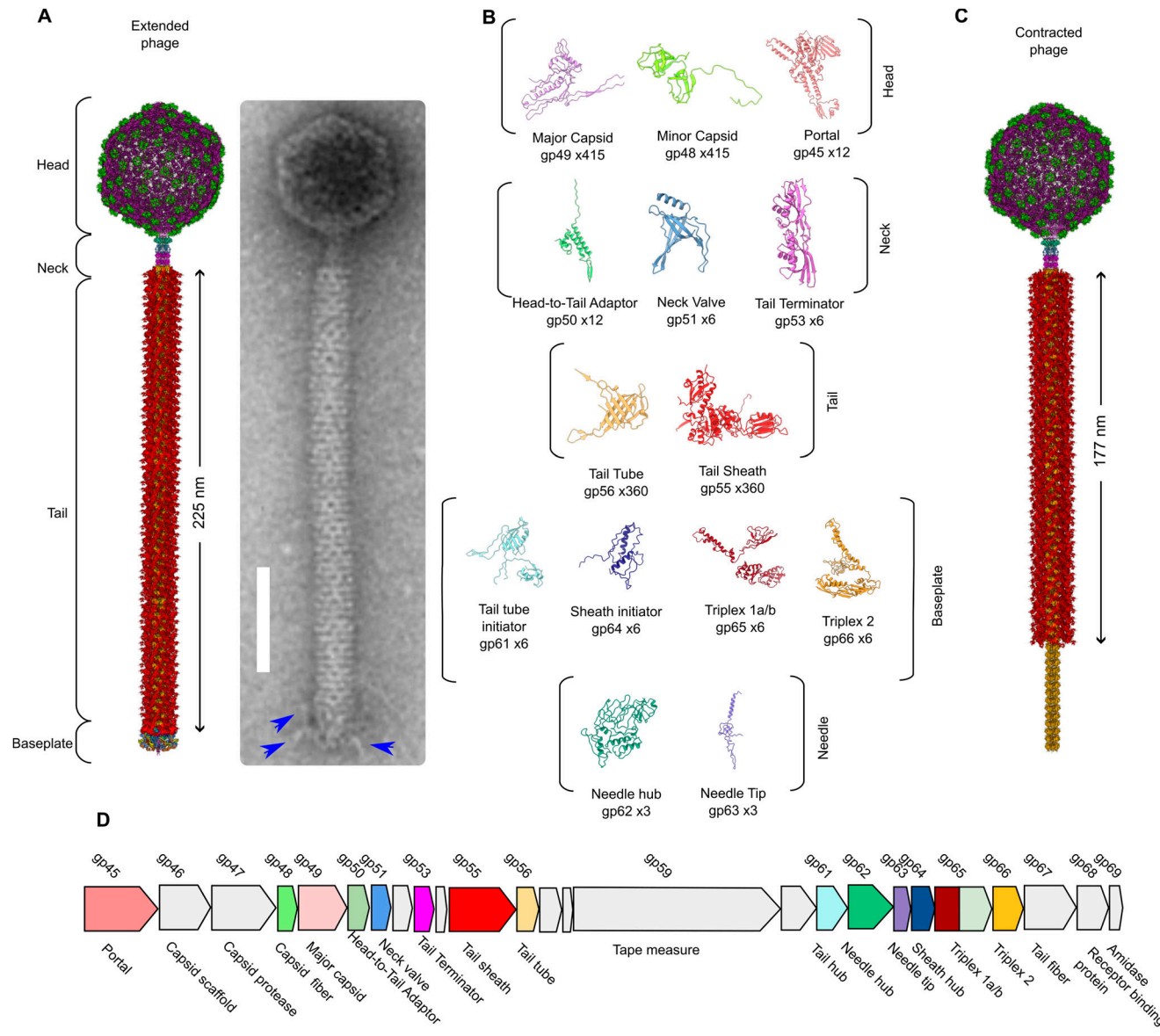

**Figure 1. Structural overview of φCD508.**
**(A)** Composite model of entire extended φCD508 phage generated from overlapping features of each model (left). Negative stain image of φCD508 with clear tail fibers attached to the baseplate (right). **(B)** Gallery of proteins built into cryoEM maps of φCD508, with protein name, gene product number, and copy number in the entire extended phage. **(C)** Composite model of entire contracted φCD508 phage generated from overlapping features of each model. **(D)** Genome organization of the φCD508 structural cassette consisting of gene products 45–68. Proteins built into cryoEM maps are colored as in the gallery, and proteins, where no model could be built, are colored gray.

## Tail

The tail is attached to the neck and made up of a stack of 57 nested hexameric tail protein rings, with an inner ring of the tail tube protein gp56 and an outer ring of the tail sheath protein gp55 (Figs 1, 2D, and 3). Each ring is offset from those above and below such that the extended tail assembly can be described as a six-stranded helix, with a twist of 19°, a rise of 39 Å, and an outer diameter of 230 Å (Figs 1B, 2D, and 3A). This arrangement of tail proteins in the extended state is similar to other myovirus and phage tail–like particles. However, a notable feature is that the tail is considerably

longer (225 nm) than those of other phages whose structures have been described (Fig 4A and B).

The main tail tube protein, gp56 (Fig S4A), has the conserved fold seen in other *Caudoviricetes* tail tubes (Arnaud et al, 2017; Zheng et al, 2017b; Kizziah et al, 2020; Zinke et al, 2020). However, gp56 lacks both the α-loop and the N-loop seen in other myovirus tail tube proteins, as well as the C-terminal loops seen in siphovirus tail tube proteins (Zinke et al, 2020) (Fig S4D). This results in a more open packing between rings of the tail tube allowing for bending of the phage in the contracted state (Figs 5 and S4B and C).

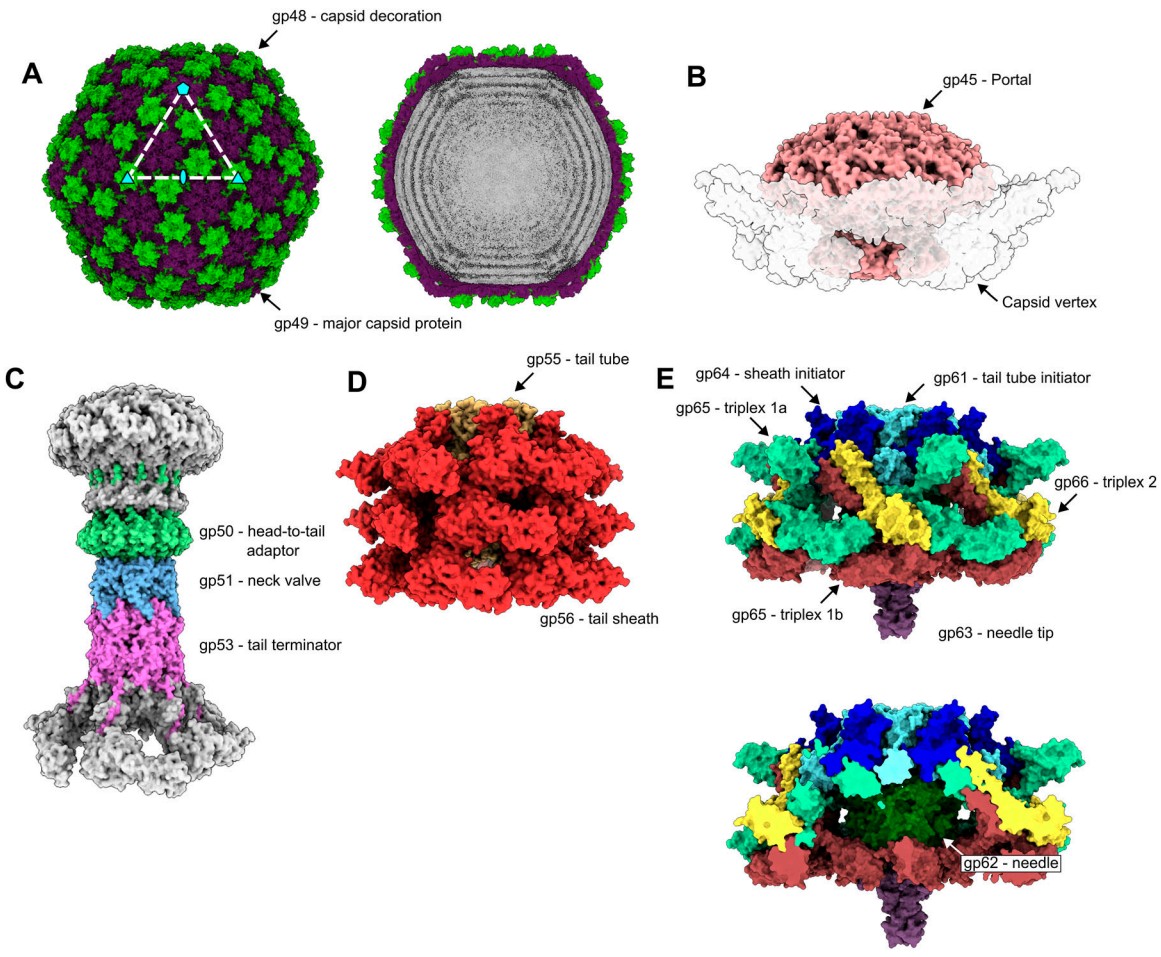

**Figure 2. Structural assemblies of φCD508.**
Shaded surface representation of φCD508 assemblies as determined by single-particle analysis. **(A)** Capsid consisting of gp48 and gp49. Slice through extended phage capsid shows DNA layers (right). **(B)** Portal protein dodecamer within the unique pentameric capsid vertex (shown with a transparent surface). **(C)** Neck proteins shaded by protein identity (head-to-tail adaptor = green, neck valve = blue, tail terminator = pink), with portal and sheath proteins shown in gray for context. **(D)** Three layers of sheath (red) and tail tube protein (orange) in the extended state. **(E)** Baseplate and needle assembly with hub (blue and cyan), wedge (yellow, mint, and brown), and needle (green and purple).

The gp55 tail sheath protein is made up of three domains (Figs 3C and S5). The tail tube–proximal domain I contains a loop consisting of residues 368–378, which is twice as long in φCD508 than in the sheath domains of pyocins, AFP, and PVC (six residues and three residues, respectively) (Heymann et al, 2013; Ge et al, 2015; Desfosses et al, 2019; Jiang et al, 2019). For a typical "full" contraction, this loop ("X-loop") would interact with the linker of the N-terminal β-strand insertion of a neighboring sheath protein (Fig 3B and C), significantly reducing the possible range of motion of the linker compared with other known CIS structures (Fig S4E and F).

## Baseplate and needle

The tail is terminated by the baseplate. Importantly, the baseplate of φCD508 must be adapted for binding and penetration of the host cell S-layer. The baseplate is minimal compared with other structurally characterized phages (Novacek et al, 2016; Taylor et al, 2016; Guerrero-Ferreira et al, 2019) consisting of only six unique

proteins (Figs 2E and S6). These are organized into a baseplate hub (Fig S7A), which connects to the tail, and a wedge, which forms a collar around the needle (Figs 2E and S7C). The baseplate is also expected to include the host attachment proteins gp67 and gp68, constituting the tail fiber and receptor binding proteins that bind to the *C. difficile* S-layer (Phetruen et al, 2022; Royer et al, 2023). Although these could not be sufficiently resolved in the EM maps for atomic modeling, low-pass–filtered maps show protrusions for the tail fibers (Fig S7F). The quaternary structures of gp67 and gp68 are unknown; however, AlphaFold3 predictions of the single polypeptide structures suggest that they both have long α-helical domains; in a higher order assembly, these helices could form bundles consistent with the extended limbs making up the attachment machinery (Figs 5A, S7F, and S8). Consistent with the formation of a gp67/68 complex, we have previously shown that swapping the C-terminal half of the gp67 paralog and the associated gp68 is sufficient to change the target range of an engineered tailocin (Gebhart et al, 2015; Kirk et al, 2017).

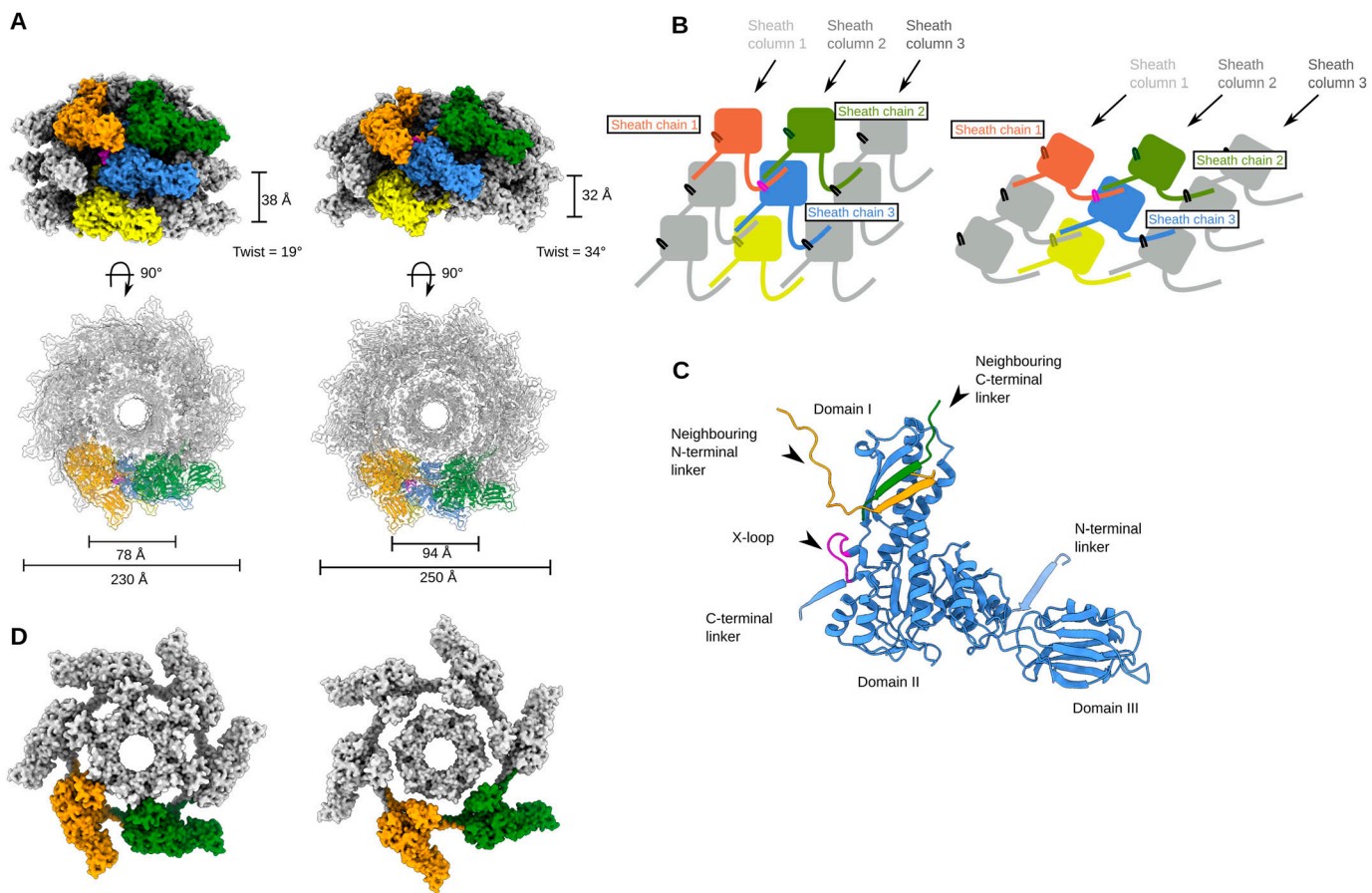

**Figure 3. Reduced contraction of the φCD508 sheath, relative to other contractile injection systems.**
**(A)** Surface rendering of the extended tail (left) and contracted tail (right), showing helical parameters of sheath proteins and inner and outer sheath dimensions. Four interwoven chains are colored (orange, green, blue, and yellow), and the X-loop for the blue chain is shown in magenta. **(A, B)** Schematic of the mesh network between sheath layers colored as in (A). **(A, C)** Cartoon representation of a single sheath protein showing domain architecture and the X-loop region, colored as in (A). Also shown are the C-terminal linker and the N-terminal linker that form the extended mesh between neighboring chains. **(A, D)** Surface rendering of the top layer of the sheath and tail tube proteins as in (A). In the extended state (left), the sheath forms extensive interactions between the sheath and the tail tube. In the contracted form (right), the modeled tail tube is able to pass through the sheath ring.

Within the ring formed by the baseplate proteins lies the needle assembly; this is the other essential component presumed to be adapted for S-layer penetration. It is formed of two proteins, gp62 and gp63, providing a pointed end terminating the tail (Figs 2E and S7B). Peptidoglycan hydrolases that facilitate enzymatic penetration of the cell wall are a common feature of phage baseplates (Arisaka et al, 2003; Taylor et al, 2016; Dunne et al, 2018; Guerrero-Ferreira et al, 2019), including in the recent structure of a diffocin, a native *C. difficile* tailocin (Cai et al, 2024). It is striking that we find no obvious auxiliary domains with structural homology to peptidoglycan hydrolases in any of the φCD508 baseplate or needle proteins built to near completion within our cryoEM maps. Furthermore, exhaustive bioinformatics analysis of all proteins encoded within the φCD508 structural cassette (including those not modeled but identified by MS-MS analysis of purified virions) identified no enzymatic domains or motifs with predicted peptidoglycan hydrolase activity (Supplemental Data 1). It is also striking that the needle spike protein lacks the β-helix observed in other CIS structures (Fig S9A), and instead, the OB-fold is connected directly to the

metal-binding apex domain (Fig S7B). A detailed comparison of these differences is explored in Supplemental Data 2.

### The φCD508 sheath does not contract as much as other CISs

The overall RMSD between all proteins of the neck in the extended state compared with the contracted state is 0.34 Å, indicating that no large conformational change occurs in this region during phage contraction. After contraction and genome release, the neck-proximal tail sheath protein (gp55) remains attached to the tail terminator gp53 via its C-terminal insertion domain (Fig S3H and I). The neck-proximal sheath protein rotates to accommodate the contraction-induced movements of the other sheath proteins in the tail below, pivoting around Gly260 in the gp53 C-terminal linker (Fig S3I and K). The terminal sheath ring proximal to the neck proteins still maintains some contact with the tail tube. This reduced rotation is limited to the proximal layer, with the next layer below fully dissociating from the tail tube, similar to the

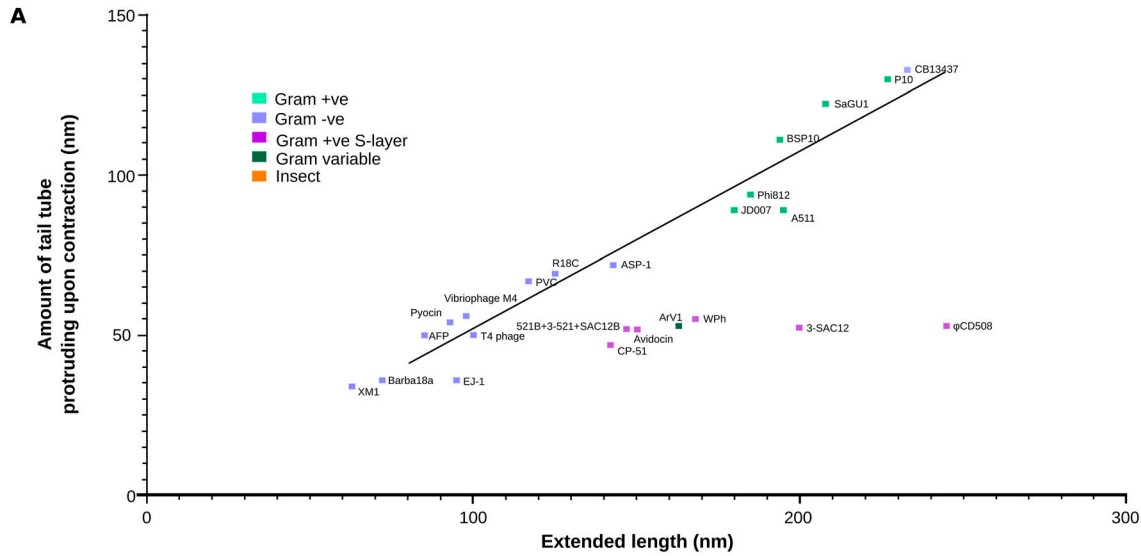

**Figure 4. Comparison of contraction parameters in CISs.**
**(A)** Graph of the extended tail length versus the amount of tail tube protruding upon contraction for a number of contractile bacteriophage and phage tail-like particles from the literature. Each particle is colored based on the bacterial/insect host type. "Conventional" virions fit well along a trendline (R2 = 0.975), whereas S-layer–penetrating phages and diffocins of Gram-positive bacteria (including CD508 in the current study) have a tail tube protrusion length independent of tail length. **(B)** Table of tail parameters for bacteriophage and phage tail–like particles for which structural data are available.

arrangement observed for the main body of the contracted tail reconstruction.

Contraction of φCD508 in urea reduces the tail length by just 20%, whereas most CIS tails that have been described contract to ~50% of their extended length; the outer diameter of the sheath increases from 230 to 250 Å, and the inner diameter from 78 to 94 Å, sufficient to break contacts between gp55 sheath domain I helices and the tail tube gp56 (Fig 3D). The contracted sheath retains the arrangement of gp55 subunits in a six-stranded helix, but with a decrease in helical rise from 39 Å to 32 Å, and a concomitant increase in twist from 19 to 34°. The contraction-induced movement of the gp55 sheath proteins can be modeled as a rigid body pivoting motion about domain I (Video 1, Video 2, Video 3, and Video 4; Fig 3A and B).

Electron cryotomography of φCD508 virions incubated with S-layer fragments was carried out once it was established that phages contracted on intact cells; this confirmed the 20% reduction

in sheath length in the presence of the natural receptor (225–177 nm) (Fig 5C). In the urea-contracted form of φCD508, the baseplate was lost, whereas the baseplate was retained when bound to the native S-layer receptor (Fig 5B and C). Notably, at intermediate time points in the incubation we found that some fully contracted virions had not yet released their genome from the capsid (Fig 5D). This was previously observed in images of the *S. aureus* phage φ812 (Novacek et al, 2016).

In both the extended and the contracted form, the sheath ring retains integrity through its mesh network, similar to the sheath of other CISs (Desfosses et al, 2019; Jiang et al, 2019; Ge et al, 2020) (Fig 3B and D; Video 1, Video 2, Video 3, and Video 4). The overall RMSD between sheath protein monomers in the extended and contracted form is 0.63 Å, confirming a rigid body movement of these proteins. In helical reconstructions of the contracted tail, the tail tube cannot be resolved because of a mismatch between the helical parameters of the sheath and the tail tube. As the tail tube does not contract,

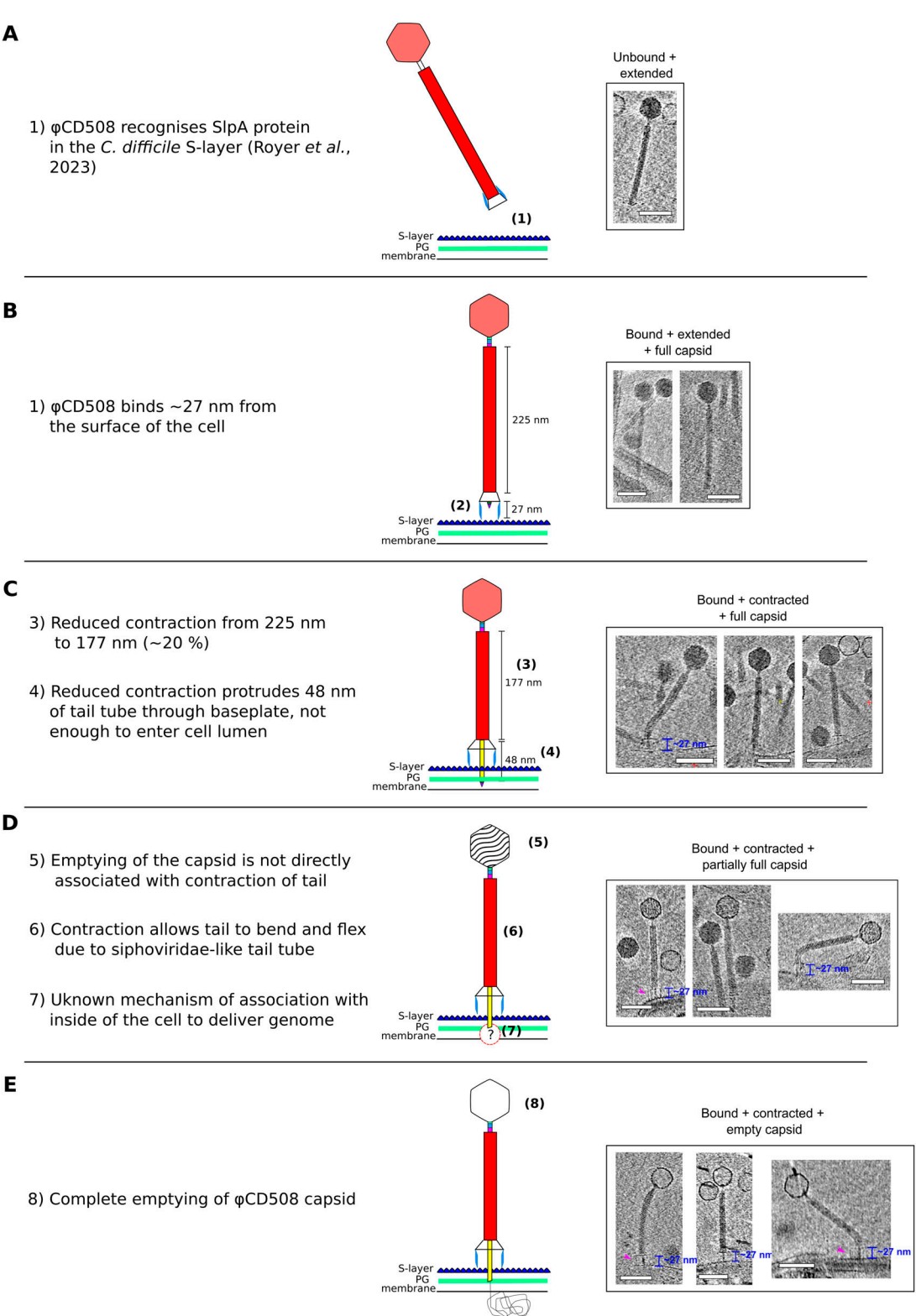

**Figure 5. Model for φCD508 reduced contraction.**
Schematic model and description of φCD508 infection, with a gallery of φCD508 bacteriophage bound to S-layer fragments from cryoelectron tomograms. The order of images follows the predicted model of phage infection, building on the known stages of phage contraction for other phages, as well as highlighting novel stages for φCD508 and speculative stages requiring more study. **(A, B, C, D, E)** These include well-characterized stages of free extended phage (A), attachment to the S-layer surface (Royer et al, 2023) (B), contraction but no DNA release from the capsid (C), partial emptying of the capsid (D), and empty capsids (E). Stages ii–v show examples where the phage is able to bend once contracted. Scale bar = 100 nm.

the resulting mismatch in length drives the tail tube through the bottom of the baseplate. The measured 20% reduction (48 nm) in tail length is consistent with a decrease in helical rise from 39 Å to 32 Å for 57 rings of sheath protein.

# Discussion

Here, we present a complete atomic model of the contractile bacteriophage, φCD508, in both extended and contracted states (Fig 1). φCD508 is notable as it is adapted for binding to and penetration of the host cell S-layer, whereas all of the phages that have been structurally characterized previously infect bacterial hosts that lack this widely encountered barrier (Novacek et al, 2016; Zheng et al, 2017b; Chen et al, 2017; Guerrero-Ferreira et al, 2019; Li et al, 2023; Wang et al, 2023). A striking difference between φCD508 and other known CIS structures is the much smaller relative contraction (20% compared with ~50%; Figs 1 and 5; Video 1, Video 2, Video 3, and Video 4) although the recently reported diffocin structure shows a relatively small 35% contraction (Cai et al, 2024). There are other phages described in the literature for which reduced contraction has been observed, but not yet analyzed (Klumpp et al, 2014; Kaliniene et al, 2017; Feyereisen et al, 2019). Remarkably, these phages also infect Gram-positive species that have S-layers. It is also notable that the tail tube protrusion length in these phages (and diffocin) seems independent of the total tail length (Fig 4A). To the best of our knowledge, for all other CISs studied in detail, the tail tube protrusion length is correlated with the total tail length (Fig 4A). In the contracted phage tail, which lacks the packing interactions between the tail tube and the sheath, the bending stiffness (persistence length) is also significantly reduced (Fig 5D and E). Compared with other known contracted CIS structures, the more open packing of subunits both within the tail tube and tail sheath (Figs 3 and S4) may allow the tail to be more flexible, similar to the intermediate states of A511 contraction where flexibility is also observed (Guerrero-Ferreira et al, 2019).

We reason that the reduced contraction of φCD508 is mediated by the sheath, as virions in which the baseplate has become detached remain in the same contracted state (Fig 5), suggesting this state is stabilized by the sheath proteins alone and not influenced by partial rearrangement in the baseplate. The N-terminal linker forms part of the mesh that connects the sheath proteins to one another, and has been shown to act as a hinge maintaining links between subunits through the sheath contraction (Video 2, Video 3, and Video 4) (Ge et al, 2015; Fraser et al, 2021). The increased length of the X-loop acts to restrict the motion of this N-terminal hinge compared with other CISs (Figs 3C and S4E). Furthermore, the X-loop blocks complete contraction of the tail for φCD508 (Fig 3); we modeled the φCD508 sheath proteins into a more contracted conformation, based on pyocin trunk structures (Ge et al, 2015), but increased contraction of φCD508 sheath proteins led to steric clashes between the X-loop and the neighboring N-terminal linker, as well as domain II of a neighboring sheath (Fig S4F).

A recent study showed that the rotation and translation of the tail sheath of pyocins occur simultaneously during contraction (Fraser et al, 2021). The reduced compression between layers of the

sheath protein in φCD508 is not associated with a reduction in rotation. The helical twist between sheath disk layers increases from 19 to 34° during contraction, similar to other CISs (Ge et al, 2015; Novacek et al, 2016; Desfosses et al, 2019; Guerrero-Ferreira et al, 2019; Jiang et al, 2019) (Fig 4B). Therefore, further contraction of φCD508 would lead to an over-rotation that cannot be accommodated by the mesh-like interactions between sheath proteins.

The relatively fixed tail tube protrusion length, independent of the overall tail length, in phages that infect S-layer–coated bacteria is striking (Fig 4A); this includes the phage tail–like diffocins that also target *C. difficile* strains and that have shorter tails than φCD508 (Gebhart et al, 2012; Cai et al, 2024). This would imply that the change in helical rise per subunit upon sheath contraction varies widely and may reflect the different energetics involved in penetrating the very diverse types of S-layers and cell envelopes found in different species (Fagan & Fairweather, 2014).

Uniquely, here we were able to visualize the exposure of the tail tube in the context of S-layer penetration. The major S-layer protein, SlpA, acts as the φCD508 receptor; we have previously shown that sequence variation within the main S-layer subunit SlpA (SLCT) correlates with phage infection spectrum and that the heterologous expression of the SLCT-10 SlpA alone was sufficient to sensitize a normally resistant *C. difficile* strain to φCD508 (Royer et al, 2023). Tomograms of φCD508 bound to S-layer fragments (Fig 5) show how the phage interacts with the host during infection. Side views of φCD508 bound to the fragments show that the baseplate is held ~27 nm from the S-layer surface by the tail fibers (Fig 5C–E). The thickness of the *C. difficile* cellular envelope is ~37 nm, and so combined with the ~27-nm distance from the S-layer, 48 nm of the exposed tail tube would appear to be insufficient to reach the cell membrane and enter the cytoplasm. In T4 phage infection, the membrane is actively pinched outward by 16 nm normal to its plane to meet the tail tube. It has been proposed that the tape measure protein gp29, containing a hydrophobic stretch, plays a role in this pinching to form a channel through which DNA is injected (Hu et al, 2015); it is notable that the φCD508 tape measure protein gp59 has a predicted transmembrane domain of three α-helices and may play a similar role to T4 gp29 (Fig S8). We also imaged phages contracted on whole cells; although the cells were too thick to allow visualization of the membrane state upon phage adsorption and contraction, we did not observe any additional contraction compared with phages contracted on S-layer fragments.

The terminus of the φCD508 needle is an apex domain as reported in other phages (Browning et al, 2012); it displays a conserved hexahistidine motif coordinating a putative iron ion (Figs S7B and S10). However, the notable feature of φCD508 is that the needle lacks the otherwise widely conserved β-helix (Fig S9A). It is currently unknown as to whether there is an advantage or requirement for this compact needle, but homology searches for other needle spikes lacking β-helix domains find widespread examples in phages infecting distantly related S-layer–producing bacteria.

The detailed mechanism by which the potential energy of the extended state is converted into work to penetrate the host envelope is unknown for most CISs. Imaging of contraction intermediates of phages indicates that contraction is triggered at the baseplate and propagates as a wave along the tail (Donelli et al,

1972; Moody, 1973; Guerrero-Ferreira et al, 2019). It has been proposed that long-lived intermediates represent a stage in infection of Gram-positive bacteria at which digestion of the cell wall is carried out by enzymes located at the tail tube spike (Guerrero-Ferreira et al, 2019). In φCD508, we do not observe any such intermediates, consistent with the hypothesis that penetration of the S-layer and cell wall does not involve any enzymatic activity. The apparent absence of peptidoglycan hydrolases has also been noted in a few lactococcal phages that infect their host only during the exponential phase, perhaps because the cell wall has not fully matured at this stage (Bebeacua et al, 2013; Stockdale et al, 2013). Whether a similar restriction applies to φCD508 remains to be determined.

We considered the possibility that the tape measure protein, gp59, might refold part of its chain into catalytic domains upon release. Using a combination of Phyre2 (Kelley et al, 2015) and AlphaFold3 (Abramson et al, 2024), we predict the tape measure to be largely α-helical with a similar predicted topology to that of other CISs (Cai et al, 2024) (Fig S8). Some segments are predicted to fold into globular domains of short, supercoiled helices; we found no matching folds in the PDB using Foldseek (van Kempen et al, 2024), but we did find matches to a range of AlphaFold predicted structures of other phage proteins, some of which have been described in Cai et al (2024). These domains are of unknown function, but do not appear to be specifically associated with any particular host envelope/cell wall type (Cai et al, 2024).

Fraser et al modeled and experimentally measured the energetics and forces generated through the contraction of an R-type pyocin (Fraser et al, 2021). Interfacial sheath–sheath subunit interactions dominated the energetics; the free energy difference between the extended and contracted states was found to be largely enthalpic. The model suggests that because of the wavelike nature of the contraction, CISs that are longer than the observed "contraction wavelength" generate similar forces. This raises the question of why the tail of φCD508 is relatively long and why, although the sheath–sheath rearrangement is not as extensive as that in other CISs, sufficient force is still generated. T6SS machines, which appear to use only mechanical force to penetrate a variety of prokaryotic and eukaryotic target cell envelopes, have extremely long tails up to ~1 $\mu$m in length. It has been argued that several tens of thousands of kcal/mol free energy could be released upon contraction (Brackmann et al, 2017; Wang et al, 2017). However, these estimates are based on changes in buried surface area of tail subunits and would not include any entropic contributions to the overall free energy change. In the case of φCD508, there is a decrease in the buried surface area of the sheath subunit upon contraction, when loss of contact with the tube is taken into account (~5,290 $\mathring{A}^2$ to ~4,990 $\mathring{A}^2$); if only sheath interfaces are considered, there is a small increase in buried surface (~4,870 $\mathring{A}^2$ to ~4,990 $\mathring{A}^2$). Thus, there must be other factors contributing to the free energy of contraction; the increased flexibility of the tail tube could be one factor, but a full analysis of the energetics and forces involved in contraction requires further study.

It is clear that φCD508 is a member of a novel class of contractile tail phages with reduced contraction (Fig 4A and B), including phages infecting other S-layer–producing bacteria. Our cryoEM structures show that although many common features are shared between φCD508 and other CIS structures, there are some very significant differences. To develop phages as novel antimicrobials, and to engineer phages targeting clinically important strains of *C. difficile*, characterization of the advantage these adaptations confer and the detailed interactions with SlpA will be vital.

# Materials and Methods

## Phage production and purification

φCD508 phages were propagated in *C. difficile* strain CD117. Cells from overnight culture were inoculated into 200 ml TY media and grown to an $OD_{600nm}$ of 0.1, before the addition of 10 mM $MgCl_2$, 10 mM $CaCl_2$, and φCD508 stock to an MOI of <0.1. Cells were grown for ~4 h, by which time most of the cells were lysed by the phage. The cells were then harvested at 4,000$g$, and the supernatant was filtered through 0.45-$\mu$m filters.

The filtered supernatant was centrifuged at 69,000$g$ for 2 h at 4°C to pellet the phage and then resuspended in 2 ml 50 mM Hepes, pH 7.4, 10 mM $CaCl_2$, 10 mM $MgCl_2$, and 10 mM NaCl, overnight with rotation at 4°C. The resuspended phages were mixed with 1 ml chloroform and centrifuged at 3,000$g$ for 10 min. The soluble fractions were overlaid on a CsCl gradient with 1 ml each of 1.45 g/ml, 1.5 g/ml, and 1.7 g/ml CsCl. These were centrifuged at 152,000$g$ for 4 h at 15°C. A white band corresponding to φCD508 was extracted by syringe and dialyzed against 500 ml 50 mM Hepes, pH 7.5, 10 mM NaCl, 10 mM $MgCl_2$, and 10 mM $CaCl_2$ overnight at 4°C. Samples were stored at 4°C.

## Phage–urea contraction

500 $\mu$l φCD508 at 6 mg/ml was diluted to 10 ml in filtered 50 mM Hepes, pH 7.4, 10 mM $CaCl_2$, 10 mM $MgCl_2$, 10 mM NaCl, 3 M urea and incubated at 4°C for 2 h. 30 $\mu$g/ml DNase I was added and incubated at 17°C for 10 min, made up to 30 ml in 50 mM Hepes, pH 7.4, 10 mM $CaCl_2$, 10 mM $MgCl_2$, 10 mM NaCl, and centrifuged at 74,500$g$ for 1 h at 4°C. The pellet was resuspended in 70 $\mu$l Milli-Q water; EM grids were prepared immediately.

## CryoEM data collection and image processing

Extended phages at 2 mg/ml were vitrified by double blotting onto Quantifoil R2/2 grids that had been glow-discharged (Cressington 208C; Cressington) for 10 s. Contracted phages at 1.4 mg/ml were vitrified by single blotting onto Quantifoil R2/2 grids that had been glow-discharged for 14 s. For both datasets, micrographs were collected on FEI Krios Titan operated at 300 kV and fitted with a Gatan K3 camera. A nominal magnification of 81,000x was used in super-resolution mode, with a super-resolution pixel size of 0.58 $\mathring{A}$/pixel. 14,533 movies were collected for extended phage, and 9,557 movies were collected for contracted phage, each containing 40 frames and with a total dose of 42 e/$\mathring{A}^2$. Raw movies were motion-corrected using MotionCor2 (Zheng et al, 2017a) with 2x binning, to give a nominal pixel size of 1.06 $\mathring{A}$/pixel, and defocus values were estimated using ctffind 4.1 (Rohou & Grigorieff, 2015).

## Helical reconstruction of tail tube and sheath in extended conformation

Extended phage tails were picked automatically using crYOLO (Wagner et al, 2019). 344 helical segments were picked from 64 extended phage micrographs using an EMAN helix boxer (Tang et al, 2007) and then used to train the crYOLO picking model. The model was used to pick 123,058 helical segments from 14,533 micrographs with a picking threshold of 0.15. Helix start and end coordinates were used to extract 500-pixel boxes in RELION (He & Scheres, 2017), with a helical rise of 40 Å. Helical rise and twist were estimated by first performing 3D reconstruction with no helical symmetry. The resulting map went to 7.3 Å resolution, with clear secondary structure features present. This was used to ascertain helical parameters, and helical reconstruction was then performed in cryoSPARC (Punjani et al, 2017), converging at 2.7 Å resolution with a helical twist of 19.2° and a rise of 38.9 Å.

Contracted phage tails were picked using a cryoSPARC filament tracer, with a filament diameter of 210 Å and a separation distance between segments of 0.5 diameters to prevent overlapping segments. 151,339 particles were extracted with a box size of 580 pixels, and 2D classification was used to select good segments, with 100 classes. 17 good classes containing 128,963 particles were re-extracted with a 400-pixel box size, and used for ab initio reconstruction in C1. Helical parameters were estimated from a homogeneous refinement as 33.9° twist and 31.6 Å rise, which were subsequently used in helix refinement enforcing C6 symmetry and using nonuniform refinement. Refinement converged at 4.2 Å resolution with a 34.0° twist and a 31.8 Å rise.

## Capsid reconstruction

Extended phage capsids were picked automatically using crYOLO, after training on 30 micrographs. 2D classification was used to select for classes with high-resolution secondary structure features, and capsids that contained DNA. ~70K particles were selected, and the coordinates were used to extract 1,000-pixel boxes in RELION. An initial model was generated by ab initio reconstruction in cryoSPARC, enforcing I1 symmetry. The initial model was then used in 3D refinement with I1 symmetry.

Contracted phage capsids were picked using RELION autopicker with a 0.4 picking threshold. 42,410 particle coordinates were used to extract particles with 1,000-pixel box size. Ab initio reconstruction was performed with 10,000 particles and enforced C5 symmetry, and used as a reference in homogeneous refinement with I1 symmetry. Refinement converged at 3.3 Å resolution.

## Baseplate reconstruction

Baseplates from the extended phage dataset were picked using crYOLO after training on a subset of 139 particles from 56 micrographs; 23,587 particles were extracted with 500-pixel box sizes and were subjected to 25 iterations of 2D classification in RELION with 30 classes and 480 Å mask. Good classes containing secondary structural features for the baseplate were selected, and 4,281 particles from these classes were used for an initial reconstruction. 50 equally spaced templates were generated from the initial

reconstruction in cryoSPARC and used to pick the final particle set. 2D classes containing baseplates were selected, incorporating 19,276 particles. A C6 ab initio reconstruction was used as a reference for homogeneous refinement imposing C6 symmetry, followed by local CTF refinement and nonuniform refinement. The final reconstruction converged at 3.4 Å and was further sharpened with a B-factor of −64.7 for inspection and model building.

## Portal and neck reconstruction

In order to locate the extended portal and neck vertex, an I4 icosahedral capsid reconstruction was generated in RELION to align a fivefold vertex along the z-axis. These I4 aligned particles were symmetry-expanded using relion_particle_symmetry_expand --i run_data.star --sym I4 --o expanded_particles.star. Expanded particles were then re-extracted with a 1,000-pixel box size rescaled to 126 pixels and then used in focused 3D classification with a tight cylindrical mask and a T-number (tau fudge) of 20. Two classes containing 230,382 particles had density for the neck and portal, and were re-extracted with recentering offset of 37 pixels in the z-direction, to center on the portal vertex. Duplicate particles were removed from the recentered particles, with a minimum distance cutoff of 108 Å. 70,554 remaining particles were re-extracted with a box size of 300 pixels and run through 2D classification in cryoSPARC. 38,968 particles containing clear secondary structure features were used in ab initio reconstruction with C12 symmetry, followed by refinement in cryoSPARC, again with C12 symmetry, and with 3 final passes after convergence; this improved the quality of the density, converging at 3.3 Å. Local CTF refinement was run in cryoSPARC, and homogeneous refinement was run with the CTF-refined particles, converging at 2.6 Å.

The extended phage neck was reconstructed in a similar way to the portal, by re-extracting the classes containing portal density from focused 3D classification, but with an offset of 50 pixels in the z-direction, which centered the neck proteins in the extracted boxes, and applying C6 symmetry in both ab initio reconstruction and homogeneous refinement of the neck in cryoSPARC. Refinement converged at 3.8 Å, and then 3.4 Å after local CTF refinement.

Contracted phage portals and necks were generated in the same way. An I4 capsid reconstruction from 33,824 autopicked capsids was symmetry-expanded, and 1,000-pixel boxes were rescaled to 126 pixels. 3D classification was run with 10 classes, a tight cylindrical mask used for the extended portal localization, and a T-number of 20. One class containing 116,487 particles was extracted with an offset of 37 pixels, and duplicates were removed with a minimum overlap distance of 60 Å, leaving 31,079 particles. These particles were extracted with a box size of 300 pixels and were imported to cryoSPARC. 2D classification was run with 50 classes, and 12 classes containing clear portal density were selected, containing 26,404 particles. The C12 reconstruction was generated using an ab initio model with enforced C12 symmetry, and converged at 2.9 Å; the C5 portal reconstruction was generated using the C12 ab initio model but enforcing C5 symmetry during homogeneous refinement, converging at 3.5 Å.

Overlapping features in the C12 portal and C6 neck reconstructions were used to generate the composite atomic model after building.

## Portal/capsid mismatch reconstruction

Firstly, aligned particles from the C12 portal reconstruction were used in homogeneous refinement in cryoSPARC using C5 symmetry, with an enforced C5 ab initio model as a reference. This reconstruction resolved the capsid density surrounding the portal complex. Next, the resulting particles and orientations were exported to RELION using the UCSF pyem routine csparc2star.py (Asarnow et al, 2019), and symmetry-expanded with C5 symmetry. The expanded particle sets were used in 3D classification with a mask around the portal, without alignment and with no symmetry imposed. Classification resulted in five classes containing ~20% of particles in each class. One class was chosen and further refined by 3D refinement. A mask was generated using Chimera's segment function to mask the capsid and portal complex, for masked refinement and post-processing (Pettersen et al, 2021).

## Model building, refinement, and validation

Purified phages were analyzed by LC-MS/MS, and identified structural proteins were built ab initio where the resolution was sufficient, or first modeled by RoseTTAFold (Baek et al, 2021) or AlphaFold (Jumper et al, 2021), and then fitted into the maps of both the extended and the contracted phage. Model building was done ab initio using Coot (Emsley & Cowtan, 2004) where the resolution was permitted. A poly-Ala trace was generated, and then, side chains were added in Phenix (Liebschner et al, 2019) using sequence from the map, with candidates from mass spectrometry data and genome data to guide identification. Where matches were found, the sequence was completed and further refined in Phenix with real space refinement, while manually correcting errors with ISOLDE's model rebuilding tools (Croll, 2018). Validation was performed using MolProbity (Williams et al, 2018).

Density for the distal domain of the sheath protein gp55 was too poor to build ab initio, and so the gp55 structure was also predicted by Robetta (Yang et al, 2020) and compared with the initially built proximal domains; a fit of 0.9 Å RMSD was achieved. The predicted distal domain was fitted with restrained flexible fitting into the map using ISOLDE.

Baseplate proteins gp65 and gp66 were also built partially from AlphaFold predicted structures using the ColabFold build of AlphaFold (Mirdita et al, 2022). Models were first built ab initio through interpretable density, followed by fitting of predicted structures and flexible fitting of the remaining domains. Completed models were run through ISOLDE with restrained flexible fitting to fit the main chain and side chains into maps of different sharpening B-factors.

## φCD508 ghost tomography

*C. difficile* strain CD117 S-layer ghosts were generated as described previously (Lanzoni-Mangutchi et al, 2022). Briefly, 80 ml TY was inoculated with CD117 to 0.05 OD and grown to OD 0.7. Cultures were centrifuged at 4°C for 15 min at 2,000*g*, then resuspended in 30 ml ice-cold deionized water. This was repeated with a resuspension in 15 ml. This was added to 15 ml precooled 212–300 $\mu$m acid-washed glass beads (Sigma-Aldrich) on ice in a homogenization flask and homogenized for 30s, then placed on ice for 5 min. The supernatant was decanted and centrifuged at 800*g* for 10 min at 4°C, and the supernatant was again centrifuged at 3,000*g* for 10 min at 4°C. The pellet was resuspended in 400 $\mu$l ice-cold 1M NaCl and centrifuged for 10 min at 4°C. Finally, the pellet was resuspended in 200 $\mu$l phage buffer (50 mM NaCl) and flash-frozen in liquid nitrogen in PCR tubes in 25 $\mu$l aliquots. 2.7 mg/ml CD508 and S-layer suspension were mixed 1:1 (vol:vol), and incubated at 37°C for 5 or 30 min and then placed on ice. 8 $\mu$l BSA-treated gold fiducials (Iancu et al, 2006) were mixed with 20 $\mu$l phage/S-layer mix, and 3.5 $\mu$l sample was applied to R3.5/1 Quantifoil grids that had been glow-discharged for 14 s. The sample was adsorbed for 1 min and then blotted for 4 s using a Leica GP2 blotting device before plunging into liquid ethane.

Tomograms were collected with Tomo software on a Tecnai Arctica 200 kV microscope fitted with a Falcon III detector. Tilt series were collected from −60° to +60° with a 3° tilt increment, starting from 20° and sweeping through to −60° before completing from 20° to 60°. At each tilt angle, images were collected with 1.2 e/Å2 over 10 frames. Tomograms were collected with an applied defocus ranging from −3 to −8 $\mu$m. Tomograms were reconstructed using IMOD using gold beads as fiducials for generating an alignment model (Kremer et al, 1996). Tomograms were reconstructed with weighted back projection, and with 15 iterations of SIRT-like filtering and 2–3 x binning.

## φCD508 tomogram tail measurements

Reconstructed tomograms were used to measure tail lengths in IMOD by selecting the neck-proximal and baseplate-proximal sheath layers, and using analyze tubes in IMOD to calculate lengths. Virions with full capsids, partially emptied capsids, and emptied capsids were measured and used to determine average lengths.

## Phage genome sequencing

Genomic DNA from φCD508 was extracted by phenol–chloroform as described previously (Sekulovic et al, 2014), then further purified using Agencourt AMPure XP magnetic beads (Beckman Coulter). Phage DNA libraries were prepared for Illumina sequencing using NEBNext Ultra II FS DNA Library Prep Kit for Illumina (New England BioLabs) according to the manufacturer's instructions. Sequencing was performed on an Illumina NextSeq 500 sequencer (paired end 2 × 75 bp) at the RNomic Platform of the Université de Sherbrooke (Sherbrooke, Québec, Canada). Trimming of the raw reads was performed using fastp (v.0.20.0) (Chen et al, 2018). Clean reads were then assembled de novo using SPAdes (v.3.14.0) (Bankevich et al, 2012) with default options. A single contig of 49,272 bp with a coverage of 1,232x was obtained.

## Genome annotation

Genome annotation was performed using Prokka (v.1.14.6) (Seemann, 2014) run locally with the vCONTACT2 protein sequence database downloaded on March 31, 2023, using the INPHARED Perl script (Cook et al, 2021). The e-value threshold was set at 1 × 10$^{-3}$. Structural proteins for which a function could not be assigned using Prokka but

that were detected by mass spectrometry were annotated based on the function predicted from the structure reconstruction. The genomic map was created using Benchling v.2.1.2 and finalized with Inkscape v.1.2.1.

## Data Availability

CryoEM maps have been deposited with the Electron Microscopy Data Bank and coordinates with the Protein Data Bank: Extended Phage Capsid I1 (EMD-51191, PDB 9GAY), Extended Phage Portal C5 (EMD-51193, PDB 9GB0), Extended Phage Portal C12 (EMD-51196, PDB 9GB3), Extended Phage Neck (EMD-51200, PDB 9GB7), Extended Phage Tail (EMD-51194, PDB 9GB1), Extended Phage Baseplate (EMD-51195, PDB 9GB2), Extended Phage Needle (EMD-51138, PDB 9G8S), Contracted Phage Capsid I1 (EMD-51192, PDB 9GAZ), Contracted Phage Portal C12 (EMD-51197, PDB 9GB4), Contracted Phage Neck (EMD-51198, PDB 9GB5), Contracted Phage Tail (EMD-51199, PDB 9GB6). The complete φCD508 genome sequence is available through the National Center for Biotechnology Information database under the accession number OR295560. Requests for materials should be addressed to PA Bullough (email:p.bullough@sheffield.ac.uk) or RP Fagan (email:r.fagan@sheffield.ac.uk).

## Supplementary Information

## Acknowledgements

We thank Svetomir Tzokov of the School of Biosciences Electron Microscopy Facility, University of Sheffield, for assistance with EM and Adelina Acosta Martin of the Faculty of Science Mass Spectrometry Facility for assistance with MS analysis. We acknowledge Diamond Light Source for access and support of the cryoEM facilities at the UK National Electron Bio-Imaging Centre (eBIC), Yun Song and Vinod Kumar Vogirala for help with data collection, and Mathew Arnold for help with data processing. We thank Alexia Royer for communicating preliminary data on phage infectivity. We are most grateful for helpful discussions with Rebekkah Menday, David Rice, Rebecca Corrigan, and Indrajit Lahiri. This work was supported by the University of Sheffield's Imagine:Imaging Life Programme, the Biotechnology and Biological Sciences Research Council grant BB/P02002X/1 (PA Bullough and RP Fagan), and UK National Electron Bio-Imaging Centre (eBIC) proposal EM19832, funded by the Wellcome Trust, Medical Research Council, and Biotechnology and Biological Sciences Research Council. For the purpose of open access, the authors have applied a Creative Commons Attribution (CC BY) license to any Author Accepted Manuscript version arising.

### Author Contributions

JS Wilson: data curation, formal analysis, validation, investigation, visualization, methodology, and writing—original draft, review, and editing.
L-C Fortier: resources, methodology, and writing—original draft, review, and editing.
RP Fagan: conceptualization, formal analysis, supervision, funding acquisition, visualization, methodology, project administration, and writing—original draft, review, and editing.
PA Bullough: conceptualization, data curation, formal analysis, supervision, validation, visualization, methodology, project administration, and writing—original draft, review, and editing.

### Conflict of Interest Statement

The authors declare that they have no conflict of interest.

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
