## [Reviewer comments · Life Science Alliance]

Life Science Alliance

Molecular mechanism of bacteriophage contraction-structure of an S-layer-penetrating bacteriophage

Jason Wilson, Louis-Charles Fortier, Robert Fagan, and Per Bullough

DOI: <https://doi.org/10.26508/lsa.202403088>

Corresponding author(s): Robert Fagan, University of Sheffield and Jason Wilson, University of Sheffield

Review Timeline:

Submission Date:	2024-10-11
Editorial Decision:	2024-12-03
Revision Received:	2025-02-12
Editorial Decision:	2025-02-25
Revision Received:	2025-03-03
Accepted:	2025-03-04

Transaction Report:

December 3, 2024

Re: Life Science Alliance manuscript #LSA-2024-03088-T

Dr. Robert P Fagan
University of Sheffield
School of Biosciences
Firth Court
Western Bank
Sheffield S10 2TN
United Kingdom

Dear Dr. Fagan,

Thank you for submitting your manuscript entitled "Molecular mechanism of bacteriophage tail contraction- structure of an S-layer-penetrating bacteriophage" to Life Science Alliance. The manuscript was assessed by expert reviewers, whose comments are appended to this letter. We invite you to submit a revised manuscript addressing the Reviewer comments.

Thank you for this interesting contribution to Life Science Alliance. We are looking forward to receiving your revised manuscript.

Sincerely,

B. MANUSCRIPT ORGANIZATION AND FORMATTING:

Reviewer #1 (Comments to the Authors (Required)):

This is a generally well-written paper that describes the architecture of bacteriophage ϕ CD508, a myoviridae that infects the Gram-positive human pathogen *Clostridioides difficile*. The cryo-EM work is rigorous, but the text is a bit dry, and, at times difficult to follow. The authors take an inventory of all ϕ CD508 structural components with limited narrative or connectivity among different parts. Most readers would lose interest in this work after a few lines, although the structures are indeed important and somewhat novel.

Major points:

- 1 - Introduction: the paper provides minimal information about phage ϕ CD508 biology. Is this phage lysogenic? If so, the discussion of phage therapy should be tuned down. Also, I couldn't find genome assembly for this phage in NCBI despite an accession number being provided. Is ϕ CD508 genetically similar to other known phages?
- 2 - Sheath contraction mechanism. It is not clear how the sheath protein undergoes conformational change in this phage. Figure 3 provides no immediate visualization of the contraction process. Also, the text refers to 'reduced contraction', but relative to what?
- 3 - Line 190-195. "It is striking that we find no obvious auxiliary domains with structural homology to peptidoglycan hydrolases in any of the ϕ CD508 baseplate or needle proteins built to near-completion within our cryoEM maps. Furthermore, exhaustive bioinformatic analysis of all proteins encoded within the ϕ CD508 structural cassette, including those not modeled but identified by MS-MS analysis of purified virions, identified no enzymatic domains or motifs with predicted peptidoglycan hydrolase activity". The authors should summarize and present the Mass Spec data in the results; state how many ORFs are not 'seen' in the reconstruction and describe the putative function of the unassigned gene products.
- 4 - Tail fibers. There appear to be fibers emanating from the baseplate that interact with the s-layer. No effort is made to describe/interpret the density for the fibers, generate Alpha fold predictions and rigid dock them into the density. The fibers seem to be the most interesting/novel part of the paper and are barely discussed.
- 5 - Figure 5. The model is not very informative. Except for the 'reduced contraction' step, all other steps are either known or speculative. Also, is the membrane shown in scale with the tail? Shouldn't the S-layer be thicker?

Minor points:

- 6 - In Myoviridae the tail 'needle' is usually referred to as a 'spike' or 'tip'
- 7 - No references for Fig. 4B

Reviewer #2 (Comments to the Authors (Required)):

The manuscript "Molecular mechanism of bacteriophage tail contraction structure of an S2 layer-penetrating bacteriophage by Jason S. Wilson et al. describes, using cryoEM, the structure of a myophage infecting a gram-positive bacterium and binding/penetrating its S-layer. This is already - to my knowledge - a novel finding. The second one is that the contractile tail of the phage is very long compared to other known structures of myophages. The third novelty is that this phage does not possess enzymes to cleave the S-layer or the thick peptidoglycan (PG). I must say that I found this manuscript superb. The resolution is very good (2.6/2.9 to 4.0 Å) and the description and comparison with other structures is extensive without being boring. Obviously, there is space for minor improvements.

1/ Looking at the structure made me immediately think to T6SS. This system is often described as phage captured and tamed by gram-minus bacteria. The length of its contractile tail is up to a micro-metre. The potential energy when released is able to punch hostile bacteria without any other help. I would suggest to mention it in the introduction/discussion. There are paper(s)

analysing the energy stored in T6SS tail, this might be useful to the authors in the discussion.

2/ There are numerous structures of siphophages infecting gram+ bacteria that may provide other informations. Some lactococcal siphophages, for exemple phage p2, are totally devoid of PG cleaving enzymes. In this case, it was shown that they can infect bacteria only in the exponential and tot the stationary phase, due to the fact that there are area in the dividing bacterium in which the PG is not cross-linked yet. Any clues in the present system?

3/ A third possibility is to have the PG cleaving enzymes within the TMP. When the TMP is released from the tail, parts of it fold (after the TMH domains) in PG cleaving enzymes. A few systems are described (e.g. in strep phages) and AF3 exquisitely predicts the structures. This should be tested here.

Reviewer #3 (Comments to the Authors (Required)):

Wilson and colleagues present structural insight into the mechanism of action of a Clostridium-targeting bacteriophage (ϕ CD508), which needs to overcome the bacterial S-layer. Overall, this is a well-written manuscript describing important results that will appeal to a wider audience interested in bacteriophages. The structural work appears to be well done based on the cryo-EM and refinement, but if there is a future revision it would be appreciated if the authors can share maps and models. I think the manuscript would still benefit of some further evidence on the absence of auxiliary domains for S-layer degradation, as described below.

Major points of the paper and provided evidence:

No auxiliary domains for S-layer degradation:

Medium support. It is not really possible to prove that such domains are not there, as there might be domains lost in the purification procedure. I strongly suggest the authors perform the following experiments:

1) What is the difference in infectivity between phages from the phage stock vs phages that were prepared for cryo-EM. Are the latter even infective? Maybe the preparation procedure gets rid of some auxiliary domains/proteins. (experiment very strongly suggested).

2) Can the authors show phage injection into cells rather than isolated S-layer fragments? (perform experiment if possible)

The phage contraction is only partial:

Well supported. Could be improved if the authors show phage+cell experiments (see before).

Other than that

The abstract could be improved and made more specific, at the moment e.g. the name of the phage is not mentioned.

Line 39, authors could stand out that other CISs, despite not injecting DNA, may be able to inject proteins as some phages do as well.

Line 41, more high resolution contracted structure have been released, for example phi812 (doi: <https://doi.org/10.1101/2024.09.19.613683>).

Line 104, is the DNA ejection partial or total? Can the authors verify this or see if there are any remaining DNA rings around the portal or somewhere else in the capsid? Related to this and maybe to consider first, are the authors able to see any arrangement of the DNA around the portal on the pre-ejected state as it happens for other phages or is not visible in this case?

Line 120, the ~23 Å diameter is measured from the side chain, does the resolution of the map in this region allow confident orientation of the side chain? Also, label K53 on side view of Fig.S3E seems to be misplaced.

Line 121, how similar is this gp51 to the gp16 of SPP1? Could the authors provide some RMSD or domain conservation? There are other more recent high resolution neck structures such as phage Milano (<https://doi.org/10.1038/s42003-023-05292-1>) that also referred to 'stopper' proteins, are they similar as well?

Line 157 "surface" (start with small letter)

Fig4, Table is missing references (right column is nearly empty)

It would be interesting to present the different buried surface and energy of the interactions between the sheath conformations and compare it to other contractile tails. Since the authors propose that the contraction is enough for penetration and that there is need of high energy for it, maybe there is significant energy difference between both states to support this. This information does not appear in the supplementary either, but it is mentioned for gp50 and 56. The lack of a baseplate in the postcontracted structure may diminish this value and sequence of contraction would provide actual relevance for this, but it is understandable if it is beyond this study.

Authors mention the compactness of the needle in the introduction but does not appear again on the results. Once again, some

buried surface, interaction, molecular weight of the needle compared to other systems could provide some hints on this.

FigS5, Would gp5 of T4 not be more equivalent to the needle than the gp5.4? I am also unsure about the equivalences of gp27, 48 and 54 in this figure, how did the authors determine the equivalence? Is it based on LysM or other domain presence as mentioned in supplementary or something further? Of course T4 needle, hub and tube initiator are assembled differently, but coloring seems confusing.

Supplementary

Methods:

Phage urea contraction by direct dilution has shown to affect protein integrity of T4, losing the neck fibritins, and therefore dialysis is preferred. Could this be a reason for the loss of the baseplate (does not happen for T4 but they are considerably different baseplates)? Has GuHCl been tested for contraction? Did temperature contraction show the disassembly of the baseplate as well?

CryoSPARC: not written consistently with uppercases throughout the text.

Line 86: I symmetry: which type (mentioned in other places).

Response to reviewers

Reviewer #1

- *the text is a bit dry, and, at times difficult to follow.*

In this paper we are describing a very large and complex structure with multiple interacting polypeptides. By its nature the paper requires detailed descriptions of each component albeit that we have to consign the finer details to the supplemental material. We note that the other two referees consider the manuscript well written. Referee #2 says “*I found this manuscript superb...and the description and comparison with other structures is extensive without being boring*”; referee #3 says “*this is a well-written manuscript*”

1 - Introduction: the paper provides minimal information about phage ϕ CD508 biology. Is this phage lysogenic? If so, the discussion of phage therapy should be tuned down. Also, I couldn't find genome assembly for this phage in NCBI despite an accession number being provided. Is ϕ CD508 genetically similar to other known phages?

We have modified the second last paragraph in the introduction to provide more details of ϕ CD508 biology - it is indeed lysogenic, as are all currently known *C. difficile* phage. However cocktails of lysogenic phage have proven effective in models of CDI and there is potential for lysogenic phage to be engineered to adopt a strictly lytic lifestyle. This has also been added to the text.

The ϕ CD508 genome has been deposited under accession number OR295560 but is embargoed until publication of this manuscript. For the purposes of review we have provided a copy of the underlying data, including the genome sequence and EM maps and models, here.

2 - Sheath contraction mechanism. It is not clear how the sheath protein undergoes conformational change in this phage. Figure 3 provides no immediate visualization of the contraction process.

In order to illustrate this better we have made additional supplementary videos which illustrate the rigid body morphing from extended to contracted tail with pivoting around the terminal strands of adjacent sheath subunits- see Video 2-4

- *Also, the text refers to 'reduced contraction', but relative to what?*

We have modified the legend to Fig. 3 to explain this.

3 - Line 190-195. "It is striking that we find no obvious auxiliary domains with structural homology to peptidoglycan hydrolases in any of the ϕ CD508 baseplate or needle proteins built to near-completion within our cryoEM maps. Furthermore, exhaustive bioinformatic analysis of all proteins encoded within the ϕ CD508 structural cassette, including those not modeled but

identified by MS-MS analysis of purified virions, identified no enzymatic domains or motifs with predicted peptidoglycan hydrolase activity". The authors should summarize and present the Mass Spec data in the results; state how many ORFs are not 'seen' in the reconstruction and describe the putative function of the unassigned gene products.

The mass spectrometry data are now more fully described in the first paragraph of the results section, linking to Data S1 which includes the complete breakdown of our analysis. For proteins that were not resolved in the virion structure we had assigned putative functions only where we could be reasonably confident based on a combination of Prokka genome annotation and protein domain prediction using the full suite of InterPro tools. We have since performed AlphaFold3 modelling of all proteins encoded within the structural cassette that were not resolved in our EM structures. Foldseek analysis of those AF3 models did not shed further light on predicted protein functions.

4 - Tail fibers. There appear to be fibers emanating from the baseplate that interact with the s-layer. No effort is made to describe/interpret the density for the fibers, generate Alpha fold predictions and rigid dock them into the density. The fibers seem to be the most interesting/novel part of the paper and are barely discussed.

We have deliberately avoided speculation on the tail fibres as much of the density is missing in our reconstruction and that which is visible is of very low resolution (see Fig. S7F). However, our annotation of the structural cassette genes point to the fibres consisting of two types of polypeptide- gp67 and gp68. gp67 is likely to form the main tail fibre protein that attaches to the baseplate and the density we see in Fig. S7F represents an average of two conformations of the part of this protein proximal to the baseplate. gp68 is likely to contain the receptor binding domain and be attached to gp67 at a point distal to the baseplate, based on previous studies of switching host range of diffocins by replacing the C-terminus of the equivalent to gp67 and the entire gp68 equivalent (DOIs: 10.1128/mBio.02368-14 and 10.1126/scitranslmed.aah681). We have added reference to this at Line 193-195.

We do not know the oligomeric state of either gp67 or gp68 but have used AlphaFold3 to predict their respective monomeric structures. We now present these folds in a modified version of Fig. S9 which covers bioinformatics analysis of structural proteins that we have not been able to resolve in our reconstruction. We have also included a discussion of these in the text at Line 189-193.

Given that any more detailed analysis of the tail fibres would be very speculative at this stage we suggest that this is outside the scope of the current experimental work. However, our future plans do involve a more detailed interrogation of the RBP-SlpA interaction. Nevertheless, the reviewer might be interested to see the results of some preliminary modelling that could form a platform for future experiments- see figure below. In blue we have modeled gp67 (predicted tail fibre) with AlphaFold3 as a trimer- we speculate that this might correspond to a retracted conformation. In pink we have modelled a domain of gp67 as a trimer that might correspond to the extended conformation. We speculate that gp68 (receptor binding protein) is attached to the terminal alpha helical domain of gp67, but the density is not well resolved.

[Figure removed by editorial staff per authors' request]

We have altered the figure legend for Figure 5 to better show the speculative stages, and highlighted gaps in our knowledge for further study. We think the model is a valuable addition to the paper as it shows key differences compared with models from other phage structural papers such as A511 which has an intermediate contracted state (DOI: 10.15252/emj.201899455) and phi812 which shows enzymatic degradation of the host cell wall (DOI: 10.1101/2024.09.19.613683). The figure also summarizes our key findings and intuitively places them within the known literature.

Also, is the membrane shown in scale with the tail? Shouldn't the S-layer be thicker?

Our recent determination of the *C. difficile* S-layer structure (DOI: 10.1038/s41467-022-28196-w) shows it to be ~7nm in thickness (relatively thin compared with some other S-layers) and so our schematic figure is approximately to scale.

6 - In Myoviridae the tail 'needle' is usually referred to as a 'spike' or 'tip'

We have altered references to the needle 'tip' to the needle 'spike' to be in line with other *Myoviridae*

7 - No references for Fig. 4B

We have added the relevant reference for Fig. 4B

Reviewer #2

1 - Looking at the structure made me immediately think to T6SS. This system is often described as phage captured and tamed by gram-minus bacteria. The length of its contractile tail is up to a micro-metre. The potential energy when released is able to punch hostile bacteria without any other help. I would suggest to mention it in the introduction/discussion. There are paper(s) analysing the energy stored in T6SS tail, this might be useful to the authors in the discussion.

We note this very helpful set of points. We have now referred to T6SS machines in the Introduction at Line 51 and 64-66 and discussed the possible contribution of their long tails to the energetics of contraction in the Discussion around Line 359.

2 - There are numerous structures of siphophages infecting gram+ bacteria that may provide other informations. Some lactococcal siphophages, for example phage p2, are totally devoid of PG cleaving enzymes. In this case, it was shown that they can infect bacteria only in the exponential and not the stationary phase, due to the fact that there are areas in the dividing bacterium in which the PG is not cross-linked yet. Any clues in the present system?

We are grateful to the reviewer for making this point and now refer to it in the text at Line 340-342. Adsorption of ϕ CD508 and other *C. difficile* phages occurs on both log-phase (OD = 0.5) and stationary phase (overnight) cells, but we routinely perform most of our infection assays using log-phase cells to get more uniform bacterial lawns. That said, we have observed slightly reduced infectivity (1-log) and slightly more turbid phage plaques when infecting early stationary phase cells (OD = 1.0) vs log-phase cells. A similar observation was made when performing bacterial survival assays; overnight cells are more resistant to infection than log-phase cells, but in both cases, infection occurs. Due to technical challenges, our current infection data do not tell if phages can adsorb and inject their DNA into stationary phase cells, or if they inject their DNA once the bacterial cells on which they adsorbed resume their growth. Further experiments will be needed to confirm this; this awaits further study.

3 - A third possibility is to have the PG cleaving enzymes within the TMP. When the TMP is released from the tail, parts of it fold (after the TMH domains) in PG cleaving enzymes. A few systems are described (e.g. in strep phages) and AF3 exquisitely predicts the structures. This should be tested here.

We thank the reviewer for this excellent point. We have now conducted a more in-depth analysis of the tape measure protein using a combination of Phyre2 and AlphaFold3. The

results are summarised in the new Fig. S9 and described in Line 344-352. Interestingly, we see confidently predicted globular domains but we find no structural homology to any known experimentally determined enzyme structures. However, we find similar predicted domains in many phage tape measure proteins in the AlphaFold database; they have no predicted function.

Reviewer #3

- *if there is a future revision it would be appreciated if the authors can share maps and models.*

For the purposes of review we have provided a copy of the underlying data, including the genome sequence, EM maps, models and validation reports, here.

1 - What is the difference in infectivity between phages from the phage stock vs phages that were prepared for cryo-EM. Are the latter even infective? Maybe the preparation procedure gets rid of some auxiliary domains/proteins. (experiment very strongly suggested).

We agree with the reviewer that there is a risk that purification could result in loss of proteins essential for infection. However, our phage were routinely amplified from highly-purified cryoEM samples during the course of these experiments i.e. the cryoEM samples served as the stocks for generating new phage stocks periodically. Although we did not directly quantify infectivity throughout these experiments, crude ϕ CD508 lysates lose infectivity over time at 4°C, whereas our cryoEM samples retained the ability to robustly infect our host bacterial strain. Therefore, we are confident that the purification protocol produces viable and infective particles.

2 - Can the authors show phage injection into cells rather than isolated S-layer fragments? (perform experiment if possible)

We have previously looked at phages attached to whole cells in order to confirm our results with S-layer fragments. Due to the thickness of cells, the contrast is very poor and hindered any tomographic studies, but we have included one of our images here for illustration. The blue arrows show a contracted phage having released its DNA (empty head), as well as a phage in the pre-contracted state with a full head, Both phage particles are in agreement with our interpretation from S-layer fragment tomograms. A more detailed analysis would require specimen thinning by FIB milling which is beyond the scope of this paper. Note that the degree of contraction is the same as that observed with isolated S-layers.

[Figure removed by editorial staff per authors' request]

- *phage is not mentioned. The abstract could be improved and made more specific, at the moment e.g. the name of the phage is not mentioned.*

We have added the name of the phage and more specific details about the tail contraction in the abstract.

- *Line 39, authors could stand out that other CISs, despite not injecting DNA, may be able to inject proteins as some phages do as well.*

We have added a sentence to this effect in the introduction at Line 45-47, stating that CISs are able to inject effector proteins, and referencing the *Photorhabdus* virulence cassette and T6SS examples.

- *Line 41, more high resolution contracted structure have been released, for example phi812 (doi: <https://doi.org/10.1101/2024.09.19.613683>).*

We now refer to this paper and also a recent bacteriocin paper, in the introduction.

- *Line 104, is the DNA ejection partial or total? Can the authors verify this or see if there are any remaining DNA rings around the portal or somewhere else in the capsid? Related to this and maybe to consider first, are the authors able to see any arrangement of the DNA around the portal on the pre-ejected state as it happens for other phages or is not visible in this case?*

We are not able to see well-resolved DNA strands in any of our capsid/portal reconstructions, but can see a low resolution ring around the portal in the pre-ejected state. We also see layers in 2D classification that we attribute to packaged DNA. In the contracted state we see none of these bands, suggesting DNA ejection is full.

We have added a new summary of this information in the supplementary text highlighting these observations (Line 1128-1133).

[Figure removed by editorial staff per authors' request]

- *Line 120, the ~23 Å diameter is measured from the side chain, does the resolution of the map in this region allow confident orientation of the side chain? Also, label K53 on side view of Fig.S3E seems to be misplaced.*

The quality of the map is sufficient, as shown below for the relevant side chain Lys53. The label has been moved.

[Figure removed by editorial staff per authors' request]

- *Line 121, how similar is this gp51 to the gp16 of SPP1? Could the authors provide some RMSD or domain conservation? There are other more recent high resolution neck structures such as phage Milano (<https://doi.org/10.1038/s42003-023-05292-1>) that also referred to 'stopper' proteins, are they similar as well?*

For gp16 of SPP1 (blue in left panel below) the RMSD is ~ 8 Å for 53 out of 80 C-alpha pairs. In the light of recent structures we have searched for similar folds with the new Foldseek program; for an alignment pruning distance of 15 Å the best matches are Pam3 connector protein (~ 5 Å for 97 C-alpha pairs), E217 collar protein gp28 (~ 5 Å for 99 C-alpha pairs), XM1 head completion protein (~ 4 Å for 94 C-alpha pairs), stopper protein of gene transfer agent (~ 4 Å for 97 C-alpha pairs) and neck 2 protein gp15 of Milano (green in right hand panel below; ~ 6 Å for 93 C-alpha pairs). We have included this analysis at Line 143-147 in the revised text.

[Figure removed by editorial staff per authors' request]

- *Line 157 "surface" (start with small letter)*

We have left this as a capital to be consistent with all other figure legends.

- *Fig4, Table is missing references (right column is nearly empty)*

Apologies this was an oversight as we intended to update this figure after configuring the manuscript bibliography. This is now fixed in the current version.

- *It would be interesting to present the different buried surface and energy of the interactions between the sheath conformations and compare it to other contractile tales. Since the authors propose that the contraction is enough for penetration and that there is need of high energy for it, maybe there is significant energy difference between both states to support this. This information does not appear in the supplementary either, but it is mentioned for gp50 and 56. The lack of a baseplate in the postcontracted structure may diminish this value and sequence of contraction would provide actual relevance for this, but it is understandable if it is beyond this study.*

We have modified the paragraph in our manuscript to reflect these points and we have incorporated calculations of the relevant changes in buried surface area. In the paragraph starting at Line 354 we do discuss the energetics of tail contraction. A crude comparison of

change in buried surface area of sheath subunits shows very little difference (see modified text at Line 366-369) but this type of calculation comes with a high degree of uncertainty and may not account for all the entropic contributions to the free energy. For this reason, we have left a full exploration of the energetics and forces of contraction to a future study. Indeed, such detailed study (see for example the paper on pyocin by Fraser *et al.*; DOI: 10.1126/sciadv.abf9601) would be the subject of a lengthy manuscript by itself.

- *Authors mention the compactness of the needle in the introduction but does not appear again on the results. Once again, some buried surface, interaction, molecular weight of the needle compared to other systems could provide some hints on this.*

We do provide a detailed description of the needle in the supplementary information (heading *The ϕ CD508 needle complex lacks enzymatic domains, and the typical β -helix observed in other CIS needles*) but we agree that a reduced description would be useful in the main text. We have added a summary and referenced supplementary information and figures (Line 197-211).

- *FigS5, Would gp5 of T4 not be more equivalent to the needle than the gp5.4? I am also unsure about the equivalences of gp27, 48 and 54 in this figure, how did the authors determine the equivalence? Is it based on LysM or other domain presence as mentioned in supplementary or something further? Of course T4 needle, hub and tube initiator are assembled differently, but coloring seems confusing.*

We agree that gp5 of T4 is also equivalent to the ϕ CD508 needle in the OB fold it contains, however we were trying to emphasise the lack of beta-helix and lysozyme in our needle. We have now colored both gp5 and gp5.4 in T4 to better highlight the equivalence. The equivalences of T4 phage baseplate proteins to ϕ CD508 are based on structural elements and function, as for the other CISs. gp27 of T4 phage contains the C3 duplicated tail tube fold and lobe, similar to that of ϕ CD508 gp62 'needle hub'. gp48 is the terminal tail tube fold-containing protein and binds both gp27 and the sheath initiator-equivalent protein gp25. gp54, although unique in T4 is most similar in structure and function to the tail tube of ϕ CD508, however we have recoloured this grey to match the AFP representation.

- *Supplementary Methods: Phage urea contraction by direct dilution has shown to affect protein integrity of T4, losing the neck fibritins, and therefore dialysis is preferred. Could this be a reason for the loss of the baseplate (does not happen for T4 but they are considerably different baseplates)? Has GuHCl been tested for contraction? Did temperature contraction show the disassembly of the baseplate as well?*

We attempted the dialysis method as described by the reviewer, as we were also aware of the loss of neck fibritins phenomenon in T4. However, the dialysis method led to phages in which the tail sheath dissociated halfway along the length of the tail, and led to more aggregation of phages, and so we used the dilution method described. We have also attempted temperature-induced contraction as well as spontaneous contraction over time and each led to loss of baseplate. We hypothesise that the loss of the baseplate in the absence of the host surface is

due to reduced contacts between the baseplate proteins and sheath proteins with reduced contraction compared with other contractile phages. However, during phage infection, the baseplate would be held in place by the penetration of the tail tube into the host and the contraction of the tail against it. We have not tested GuHCl.

- *CryoSPARC: not written consistently with uppercases throughout the text.*

We have now corrected this.

- *Line 86: I symmetry: which type (mentioned in other places).*

We have updated the text to 'I1' to show which symmetry mode was used

February 25, 2025

RE: Life Science Alliance Manuscript #LSA-2024-03088-TR

Dr. Robert P Fagan
University of Sheffield
School of Biosciences
Firth Court
Western Bank
Sheffield S10 2TN
United Kingdom

Dear Dr. Fagan,

Thank you for submitting your revised manuscript entitled "Molecular mechanism of bacteriophage tail contraction- structure of an S-layer-penetrating bacteriophage". We would be happy to publish your paper in Life Science Alliance pending final revisions necessary to meet our formatting guidelines.

- please address Reviewer 1's remaining comments
- please be sure that the authorship listing and order is correct
- please add ORCID ID for the secondary corresponding author -- they should have received instructions on how to do so
- please add an Author Contributions to our system
- please label the panels in Figures 1 and 6 correctly
- please add callouts for Figures 4A-B; 5A,B,E; S1A-B; S2E,F; S3B-D,F,G,J; S4A-C; S5A-C; S6A-F; S7D, E; S8A-C; S9A-C and S10C to your main manuscript text
- the Supplementary Text section can be incorporated into the main manuscript. We do not have length restrictions

A. FINAL FILES:

B. MANUSCRIPT ORGANIZATION AND FORMATTING:

Sincerely,

Reviewer #1 (Comments to the Authors (Required)):

The paper is generally improved and provides more background information. However, one thing must be fixed. The author named gp50 'portal adaptor' and gp53 'tail adaptor'. I find this confusing and counterproductive. Gp50 is the 'head-to-tail adaptor' a phage protein universally conserved in nature that the author must not rename. Similarly, they have to come up with a better name for gp53.

March 4, 2025

RE: Life Science Alliance Manuscript #LSA-2024-03088-TRR

Dr. Robert P Fagan
University of Sheffield
School of Biosciences
Firth Court
Western Bank
Sheffield S10 2TN
United Kingdom

Dear Dr. Fagan,

Thank you for submitting your Research Article entitled "Molecular mechanism of bacteriophage contraction- structure of an S-layer-penetrating bacteriophage". It is a pleasure to let you know that your manuscript is now accepted for publication in Life Science Alliance. Congratulations on this interesting work.

DISTRIBUTION OF MATERIALS:

Again, congratulations on a very nice paper. I hope you found the review process to be constructive and are pleased with how the manuscript was handled editorially. We look forward to future exciting submissions from your lab.

Sincerely,
